# Adult and Larval Tracheal Systems Exhibit Different Molecular Architectures in *Drosophila*

**DOI:** 10.3390/ijms24065628

**Published:** 2023-03-15

**Authors:** Judith Bossen, Ruben Prange, Jan-Philip Kühle, Sven Künzel, Xiao Niu, Jörg U. Hammel, Laura Krieger, Mirjam Knop, Birte Ehrhardt, Karin Uliczka, Susanne Krauss-Etschmann, Thomas Roeder

**Affiliations:** 1Department Zoology, Molecular Physiology, Kiel University, 24118 Kiel, Germany; 2German Lung Center (DZL), Airway Research Center North (ARCN), 24118 Kiel, Germany; 3Department of Evolutionary Genetics, Max Planck Institute for Evolutionary Biology, 24306 Plön, Germany; 4Helmholtz-Zentrum Hereon, Institute of Materials Physics, 21502 Geesthacht, Germany; 5Research Center Borstel, Priority Research Area Chronic Lung Diseases, Early Life Origins of CLD, 23485 Borstel, Germany; 6Institute for Experimental Medicine, Kiel University, 24118 Kiel, Germany

**Keywords:** respiratory system, epithelial immune system, *Drosophila*, transcriptomics, adult specific, circadian rhythm

## Abstract

Knowing the molecular makeup of an organ system is required for its in-depth understanding. We analyzed the molecular repertoire of the adult tracheal system of the fruit fly *Drosophila melanogaster* using transcriptome studies to advance our knowledge of the adult insect tracheal system. Comparing this to the larval tracheal system revealed several major differences that likely influence organ function. During the transition from larval to adult tracheal system, a shift in the expression of genes responsible for the formation of cuticular structure occurs. This change in transcript composition manifests in the physical properties of cuticular structures of the adult trachea. Enhanced tonic activation of the immune system is observed in the adult trachea, which encompasses the increased expression of antimicrobial peptides. In addition, modulatory processes are conspicuous, in this case mainly by the increased expression of G protein-coupled receptors in the adult trachea. Finally, all components of a peripheral circadian clock are present in the adult tracheal system, which is not the case in the larval tracheal system. Comparative analysis of driver lines targeting the adult tracheal system revealed that even the canonical tracheal driver line *breathless* (*btl*)*-Gal4* is not able to target all parts of the adult tracheal system. Here, we have uncovered a specific transcriptome pattern of the adult tracheal system and provide this dataset as a basis for further analyses of the adult insect tracheal system.

## 1. Introduction

Respiratory organs have evolved in parallel in different groups of animals to allow optimal gas exchange. Especially those animals with the highest metabolic activities depend on efficient respiratory organs. Insects show impressive metabolic performance by using a respiratory system otherwise rarely found in the animal kingdom, the tracheae. Despite different ontogenetic origins, the tracheae of insects and the vertebrate lungs share various commonalities [1,2]. In both cases, we are dealing with blind-ended gas transport systems having a tree-like branched structure. Gas transport and gas exchange are also spatially separated in both organs, whereby the exchange of gases occurs in the terminal cells in insects [3] and the alveoli in mammals. Insect larvae show the original structure of a tracheal system. In contrast, the tracheal system in adult insects, especially in metabolically active flying insects, is much more complex and exhibits elements that are otherwise not or hardly ever found in larval tracheal systems. The more complex adult tracheal system comprises compressible air sacs and many anastomoses to allow for active breathing movement [4]. Recent studies have given us a basic understanding of how these very complex organs allow a highly efficient gas exchange that matches the metabolic needs during maximal physical activity [5,6,7,8,9]. While functional studies were mostly conducted with large insects, such studies are rare in the model insect, the fruit fly *Drosophila melanogaster*. However, most of our knowledge of tracheal molecular processes relies on studies with *Drosophila* [10,11]. Only recently, general aspects of tracheal development in other insects were elucidated, giving us information about how these large groups of insects manage tracheal organogenesis [12,13]. All these studies have led to an excellent understanding of the first steps of this complex process of organogenesis and the organization of the larval tracheal system, but this understanding of molecular processes relates almost exclusively to the larval system, while the adult tracheal system is mainly terra incognita in this respect [14]. For the larval trachea, particular attention focused on understanding the function of stem cells serving to form the pupal tracheal system [15,16,17,18]. Regarding processes in adult tracheal systems, the focus was on mechanisms that enable the supply of flight muscles with oxygen [19]. Even though larval and adult tracheae are respiratory organs of the same species, differences are evident. Larvae are developing animals, and adults no longer show this development, which should also be seen in the differential expression of genes.

It is noteworthy that there are only a few molecular genetic studies that focus on the larval tracheal system, with even less addressing the adult tracheal system. [20,21,22]. As a result, cellular and molecular mechanisms that are important for the functionality of the adult tracheal system are largely unknown. Compared to the larval tracheal system, the greater complexity of the adult tracheal system [23] should be reflected in differences in transcriptome profiles. Few studies addressed functional and structural aspects of the adult tracheal system using molecular genetic methods [24,25,26,27]. Recently, the FlyCellAtlas consortium has also addressed the single-cell analysis of adult tracheal cells [28]. Currently, however, no studies are publicly available, which could also be due to experimental difficulties in the specific isolation of tracheal cell nuclei.

Based on the already known background knowledge, we pursued in the present work the goal of better understanding the adult tracheal system of *Drosophila*. For this purpose, we performed a comprehensive transcriptome analysis of manually isolated adult tracheae and compared the data with those of larval tracheae. Substantial differences were revealed that point to different functionalities of larval and adult tracheae.

## 2. Results

### 2.1. Genes That Are Specifically Expressed in Adult Trachea

Tracheal systems of adult and larval *Drosophila* have different architectures. The simple basic structural plan realized in the larval tracheal system is supplemented in adults by additional elements that make the system far more complex. These structural elements, which are mainly found in the adult tracheal system, include air sacs and multiple anastomoses. To understand the molecular basis underlying these structural differences, we performed a comparative transcriptome analysis. For this purpose, we manually isolated both structures to represent the tracheal systems in their entirety and then subjected them to transcriptome analysis and ensured that no method-induced biases occurred (Figure 1). This comparison revealed distinct clustering of signatures obtained from tracheal cells to those obtained from whole flies, as shown by heat-map (Figure 1A) and principal component analyses (PCA) (Figure 1B). It yielded with 2733 a great number of differentially expressed genes (DEGs) that met the criteria (>1.5 fold up or down, FDR < 0.05, Appendix A). A total of 1897 of them were enriched, and 836 were depleted in the adult trachea. To verify whether our isolation of the trachea and the subsequent sample processing represents the transcriptome events in the adult trachea, we analyzed genes enriched in the adult trachea in more detail. To do this, we used the nine genes from this group with the highest transcript abundance (RPKM value) in the adult trachea (*whe*, *lcs*, *CG16826*, *Yp3*, *Yp1*, *CG45080*, *Yp2*, *Nplp2*, *CG34212*) to evaluate whether they were detectable in cells identified as tracheal cells in the FlyCellAtlas dataset. This was important because this was not necessarily expected for most of these genes. We found that in the cells characterized by *btl* expression, all nine candidates showed strong expression (Figure 1C). Moreover, we tested the validity of the approach by selecting genes specifically found in the adult trachea (Appendix A), specifically in the larval trachea (Appendix A) and those enriched in the adult trachea (vs. whole adults, Appendix A). In all cases, the predicted relative expression was reproduced by qRT-PCR.

To further pinpoint the role of the enriched DEG, a gene ontology enrichment (GO) analysis was conducted with the top 400 upregulated DEG in the adult trachea, which revealed a diverse network of involved biological processes (Figure 1D). This network showed a range of processes that were divided into five partially overlapping clusters and, so far, have not been associated with tracheal physiology. In the center of the network, where most genes are located, GO terms related to cell signaling, G-protein-coupled receptor signaling (GPCR), response to stimuli, and defense response to bacteria were found. Additionally, clusters with GO terms related to mating behavior and adhesion were enriched in the top 400 genes. GO terms related to the regulation of ion transport and regulation of membrane potential are also overrepresented in the adult respiratory system in comparison to the remaining of the animal (Figure 1D).

### 2.2. The Respiratory Systems from Adults and Larvae Are Clearly Distinct

*Drosophila* is a holometabolic insect and reorganizes its whole morphology during pupation to adjust tissue structures and function to the imago’s requirements. Therefore, also the respiratory system undergoes a dramatic remodeling process during metamorphosis [14]. To elucidate these differences at the level of transcription, we compared the transcriptomic signatures between the adult and larval trachea (Figure 2, Appendix A). This comparison revealed differential clustering of signatures obtained from adult tracheal cells to those obtained from larval tracheal cells, as shown by heat-map (Figure 2A) and principal component analyses (PCA) (Figure 2B). With 3034 DEGs (>1.5 up and down; FDR < 0.05), the adult and larval tracheal systems show significant differences in gene expression. All upregulated DEGs (2096) and all downregulated DEGs (938) were used in a GO term analysis, focusing on the biological processes (Figure 2C,D). In the center of the GO network, representing the upregulated DEGs in the adult trachea, the “response to stimulus/immune response” cluster contains most GO terms (Figure 2C). The GO term clusters “circadian rhythm” and mating behavior” with shared genes imply an enhanced role of the adult trachea in the corresponding processes. Two other large clusters, “transmembrane transport” and “ion homeostasis” share genes with the GO term clusters “synaptic signaling” and “carbohydrate homeostasis”, respectively. DEGs associated with relation to GO terms within “mitochondrial respiration” and “electron transport” are also enriched in the adult tracheal system. Another identified cluster with many associated GO terms is “metabolic biosynthetic processes “, which is flanked by clusters harboring GO terms related to carbohydrate and lipid metabolism. In the center of the GO network representing the downregulated DEGs in the adult trachea, the large cluster “regulation of development” includes most GO terms (Figure 2D). This implies an upregulation of development-related genes in the larval trachea, which is consistent with the impending remodeling process of the tissue during metamorphosis. Besides this, the clusters “adhesion”, “fluid transport” and “wound healing” are represented. Detached from these three clusters, only one other cluster with GO terms associated with “cytoskeleton” appears. All these GO terms containing DEGs downregulated in the adult trachea are enriched in the larval trachea simultaneously.

### 2.3. ECM and Chitin-Associated Genes Are Overrepresented in Larval Trachea

For a deeper analysis, we focused on genes showing clearly adult- or larval-specific signatures. In this context, genes related to extracellular matrix (ECM) or coding for cuticular and chitin-associated proteins were of particular interest (Figure 3). First, we looked at enzymes that are relevant for remodeling processes as they can degrade parts of the ECM (Figure 3A). Here, we focused on the members of the matrix metalloproteinase (MMP), the a disintegrin and metalloproteinase (ADAM), and the ADAM with thrombospondin motifs (ADAMTS) families. Whereas *Mmp2* was more specific for adult trachea, the opposite was true for its counterpart *Mmp1*. Among ADAM family members, *mmd* was specifically found in the adult trachea, while *Kul* was specific for the larval trachea. Finally, the ADAMTS members *sona* and *CG4096* were found in the larval trachea, while *stl* was specific to the adult trachea (Figure 4A).

The cuticular and chitin-associated proteins comprise all major gene families, including the larval cuticle proteins (*Lcp*), the structural constituent of the chitin-based larval cuticle (*Ccp*), the Tweedle (*Twdl*), the Obstructor (*obst*), the Imaginal disc growth factor- (*ldgf*) and the chitinase- (*Cht*) families (Figure 3B–H). For most relevant genes, we observed almost exclusive expression (more than 50 times different expression) in either larvae or adults. This was especially true for the *Lcp*s; as indicated by their name, they were found almost exclusively in the trachea of larvae (Figure 3B). Even more specific was the expression of members of the *Ccp*-family that were almost exclusively found in the larval trachea (Figure 3C). The *Tweedle* family members *TwdlF, TwdlG* and *TwdlE* were mainly expressed in the adult trachea, whereas *TwdlX* and *Twdlbeta* are specific for the larval tracheal system (Figure 3D). The obstructor family members *obst-A*, *-B* and *-E* were the most abundant ones, with significantly higher expression in the larval trachea, while *obst-F* and *-H* show a tendency for higher expression in the adult trachea (Figure 3E). For the *Idgf*-family, we observed a shift from *idgf4* in larvae to *idgf1* and *idgf5* in adults (Figure 3F). Similarly, among chitinases (*Cht*), *Cht8* and *Cht9* were specific for adults, while *Cht2, 6, 7*, and *10* were specifically expressed in larvae (Figure 3G). The *Cpr*-family has representatives that are almost exclusively present in larvae, including *Cpr67Fa1, Cpr12A, Cpr65Ay, Cpr65Av, Cpr47Ec* and *Cpr65Ax2*, to mention only those with more than 1000-fold higher abundance in larvae. On the other hand, *Cpr49Ab, Cpr72Ec, Cpr62Bb* and *Cpr47Ee,* were almost exclusively present in adults (>100-fold expression level; Figure 3H).

### 2.4. Overrepresented Processes in the Adult Tracheal System

Based on the number of DEGs associated with developmental processes, signaling, and intracellular transport, we had a closer look at the corresponding genes. Interestingly, a high number of receptor-, transporter-, and transmembrane channel-coding genes was transcribed at much higher levels in the adult trachea than in the larval trachea. Firstly, several GPCRs belonging to the biogenic amine receptor family, including all dopamine, all serotonin, the *TyrR* receptor, and all octopamine beta receptors, were significantly enriched in adult trachea compared to the larval trachea (Figure 4A). Moreover, all *GABA-B* type receptors and a huge variety of peptide receptors were specifically transcribed in the adult trachea. In the larval trachea, only a few receptors were specifically transcribed, comprising many *methuselah* receptors, the proctolin receptor, and the *ecdysone triggering hormone receptor* (*ETHR*). In addition to the specific G-protein coupled receptors, other parts of the signaling cascade were also expressed stage-specific, with the two arrestins *Arr1* and *Arr2* being the most prominent examples (Figure 4A). Secondly, we observed many ligand-gated ion channels, including almost all nicotinic ACh receptors and glutamate receptors, as well as various potassium channels. Finally, several transporters were specifically transcribed in the adult trachea, whereas only *SerT* was specific for the larval trachea.

Next, we looked at the melanization reaction, which is a very specific arm of the immune system, because associated DEGs showed marked differences between the larval and adult trachea (Figure 4B). Several genes coding for serine peptidase inhibitors (Serpins, Spn) were differentially expressed between larval and adult tracheal cells. However, downstream of the serine protease cascade, genes coding for Phenol Oxidases (*Phox*), such as *CG3505* or *PPO1* and *PPO2,* were specifically overrepresented in the adult trachea. Combined with enrichment in genes coding for proteins associated with dopamine synthesis, such as *Punch* and *Dopa decarboxylase (Ddc)* coding genes, Phox is responsible for dopamine-dependent melanin synthesis [29].

We also analyzed the expression of genes relevant to fat transport and metabolism and found enrichment in the adult trachea (Figure 5A) as the corresponding GO terms were enriched in the adult trachea compared with larva ones (Figure 2C). Here, almost all members of the maltase (*Mal*)-family are specifically present in the adult trachea (Figure 5A). Transport molecules such as the Trehalose transporters as well as the major carbohydrate-related genes *tobi, Amy-p*, and *Amy-d,* are also mainly present in the adult trachea (Figure 5A). With respect to the lipases *brummer* and *Lip4,* we observed the same type of distribution.

Genes associated with the production of mucin-like products also show a high degree of stage specificity. Here, we focus on mucin (*Muc-*), mucin related (*Mur-*), and salivary gland secretion (*Sgs-*)family members. In the adult trachea, we observed a generally higher expression of mucin genes, with *Muc68D*, *Mur29B* and *Muc68E* being the most specific ones (Figure 5B). These mentioned mucin-like genes are almost not expressed in the larval tracheal system, although they are of central importance as a lining system in other airway organs, such as the lung [30,31].

The observation that various components of the immune system were expressed at substantially higher levels in the adult trachea compared with the larval trachea was further analyzed. Here, a focus was on the immune deficiency (IMD) pathway, which is the most important arm of the epithelial immune system (Figure 6). Whereas the major intracellular parts of the signaling system are present to similar extents in both types of tracheae, especially the extracellular, regulatory components show a strongly divergent expression pattern. We observed higher expression of the amidases of the peptidoglycan recognition protein (PGRP) family *PGRP-SC2*, *PGRP-SB1*, and *PGRP-LB*, which should exert an inhibitory effect on IMD-signaling, but we also observed a strongly enhanced expression of *PGRP-SD*, which fosters binding of the peptidoglycan to the pattern recognition receptor *PGRP-LC* (Figure 6A,B) in the adult trachea. On the other hand, factors such as the Toll-receptors *Toll-4*, *Toll-7*, and *Tollo* (Toll-8) are expressed at higher levels in the larval trachea as well as *PGRP-LC*, *PGRP-LE*, and *PGRP-LA* (Figure 6A). A major outcome of activation of the immune system is the production of antimicrobials, most importantly, the production of antimicrobial peptides (AMP). Here, we observed a clear and often substantially higher expression of AMP genes in adult trachea if compared with larval ones (Figure 6C). As an example, we show the expression of the AMP *drosomycin* in the trachea of uninfected adult animals (Figure 6D).

Finally, we found that in the adult tracheal system, all main components of the circadian clock (Figure 7A) were expressed. The protein-coding genes for *timeless* (*tim*), *period* (*per*), *cryptochrome* (*cry*), *Clock* (*Clk*), and *cycle* (*cyc*) were strongly upregulated in adult trachea compared with larval ones. Whereas *cyc* was only four times more abundant in the adult trachea, the difference between the adult and the larval trachea was a hundred-fold and more for the other four members of the core clock (*tim*, *cry*, *per*, *Clk*) (Figure 7B). Using a tagged version of *tim* that is under transcriptional control of the natural *tim* promoter, we could show *tim* expression in the adult tracheal system (Figure 7C).

### 2.5. Specific Driver Lines Targeting Cell Populations of Adult Trachea

To visualize the tracheal structure of adult flies, we performed a detailed analysis of the adult tracheal system using micro-CT analyses of adult *Drosophila*. Here, we found a very fine meshwork of tracheal structures in the entire abdomen and different types of tracheal structures in the thorax and the head, where massive air sacs dominate (Figure 8A,B). We used this structural information obtained by micro-CT to test the ability of drivers of the Gal4/UAS system to specifically mark all adult tracheal components. We supplemented this study with an autofluorescence analysis. Here, larval and adult trachea show a marked autofluorescence at an excitation wavelength of 405 nm with a strong blue light emission [32,33]. We observed this fluorescence in the entire larval tracheal system (not shown) and in most parts of the adult tracheal system (Figure 8C). Furthermore, we also observed a very strong autofluorescence after excitation with 488 nm and emission as red light (Figure 8C). These signals are confined to tracheal parts in the thorax and the head (Figure 8C) and are never observed in the larval trachea. The problematic features of the commonly used *btl-Gal4* driver for analyzing the adult tracheal system are underpinned by the observation that manually isolated larval trachea show an approximately ten times higher expression of *btl* if compared with the manually isolated adult trachea (Figure 8D).

Here, we analyzed flies where GFP is expressed under the control of the *btl-Gal4* driver, which is still the only valid driver for the adult tracheal system. In sagittal sections of these flies, we observed similar structures in the abdomen, but significant differences, i.e., a lack of staining in the thorax and the head (Figure 1C). This lack of a precise match between the GFP expression driven by *btl-Gal4* and the tracheal structure led us to use a manual isolation approach of the adult tracheal system.

Based on the analyses of this project, we compared six promising Gal4 drivers, all of which are expressed in adult trachea [34]. Here, we compared *btl-Gal4* (Figure 8E–H), as a canonical driver used for trachea in general, with *emp-Gal4* (epithelial membrane protein, Figure 8I–L) and *flz-Gal4* (filzig, Figure 8M–P). Shown are sagittal sections of the entire animals (Figure 8E,I,M), as well as higher magnifications of the head (Figure 8F,J,N), the thorax (Figure 8G,K,O), and the abdomen (Figure 8H,L,P). *Btl-Gal4* labeled only parts of the tracheal system, where terminal cells and parts of the air sacs are labeled, various larger trachea are not labeled at all. *Emp-Gal4*, on the other hand, labeled the larger tracheal structures but missed terminal cells in the abdomen. *Flz-Gal4* had a generally lower level of expression, and it marked only parts of the tracheal structures. All three driver lines were not completely specific as other organs, such as the reproductive organs (*btl*) or the intestine (*emp-Gal4* and *flz-Gal4*), were also prominently labeled (Figure 8E,I,M). Three additional driver lines with known expression in the trachea showed lower expression levels in the trachea with higher expression in other organs. *Geko-Gal4* showed only expression in the smaller tracheal branches of the abdomen and expression associated with the carcass (Appendix A). *Ex-Gal4* showed only expression in the larger air sacs in the head and thorax but also expression in other abdominal organs (Appendix A). *Samuel-Gal4* showed a similar expression to *Emp-Gal4* but with a much lesser expression level in the trachea and a high expression level in several parts of the intestine (Appendix A). The *emp* gene was the only one that showed up in our expression analysis with RPKM values between 43 and 91 in the adult tracheal samples, which is much more than the *btl* expression (Figure 8D).

## 3. Discussion

We conducted this study to understand the basic features of the molecular and cellular organization of the adult trachea. Furthermore, we wanted to identify possible differences between larval and adult tracheae and correlate these with functional differences. To achieve this goal, we deliberately chose the experimental approach of manual dissection of adult tracheae to include all parts of the adult trachea in the analysis. Unfortunately, this is not the case when the canonical tracheal marker *btl* is used to isolate adult tracheal cells. For this reason, our work complements related efforts of the FlyCellAtlas consortium [28] and can serve as a valuable resource for further studies.

Most striking was the observation that the transcriptome signatures differed substantially between adult and larval trachea. These differences were expectable, as both systems, albeit sharing the same architecture, differ in their structure and presumably also in their physiology. Moreover, the adult tracheal system is far more elaborate with respect to different cell populations. Next to the conventional structuring into tube-like trunks and terminal cells, an additional regionalization in the head, thoracic, and abdominal trachea could enhance this diversity [35]. The need to adapt to different physiological situations is higher in adults, which might be the reason for the massive expansion of expression signatures associated with signaling in general, allowing it to influence the physiology of respiratory systems by either hormonal or neuronal information. This specific expression of many molecules closely linked to signal transduction suggests that adult tracheal cells are targets of complex modulations. Particularly noteworthy is the expression of many receptors (in particular from the family of G-protein coupled receptors) and of other molecules of the respective signal transduction cascades. Striking is the observation that almost the entire set of bioamine receptors of *Drosophila* is present in the adult trachea [36]. Furthermore, numerous peptide receptors from this family are also present in this tissue. Several rhodopsins are also present, which is surprising, but a few non-visual areas where rhodopsin expression is of physiological importance are known [37]. The high expression of arrestins one and two suggest that they are highly relevant in the adult trachea. Furthermore, the specific expression of several different ion channels and transporters in the adult trachea is noteworthy. Taken together, the idea that the higher complexity of the adult tracheal system demands a more elaborated signaling system is very appealing, but experimental data conforming to this scenario are currently lacking.

Equally evident were the changes in the epithelial immune system between developmental stages, which may be central to tracheal functionality. Tracheae represent the largest surfaces in insects, making them an ideal portal of entry for numerous pathogens. Therefore, these surfaces need protection by a highly effective immune system [20,38,39,40]. The melanization reaction is an aspect of the epithelial defense that is operative in the trachea of both larvae and adults [29,41,42]. However, we showed that central molecules of this immune signaling pathway were more abundant in adults than in larvae, indicating greater importance in the adult tracheal system. Concerning the immune response, the much higher tonic immune activity of the adult tracheal system was striking. Here, antimicrobial peptide genes from different functional classes show massive upregulation compared to the larval system, suggesting that the adult trachea exhibits tonic activation of epithelial immunity. Most genes encoding IMD pathway genes have similar expression levels in both tracheal systems. The expression of genes encoding extracellular PGRPs is much higher in the adult tracheal system. Three of these, namely PGRP-SC2, PGRP-SB1, and PGRP-LB, encode for amidases or predicted amidases. Amidases are enzymes that degrade peptidoglycan, which activates the IMD pathway. Thereby attenuating the activity of the IMD pathway. On the other hand, PGRP-SD is present at much higher levels in the adult trachea. It is known to be one of the major mediators of the IMD pathway by binding peptidoglycans and transporting them to the receptor [43]. The immune receptors Toll-4, -7, and -8 (Tollo) are present at higher levels in the larval trachea. Tollo, in particular, is relevant because it acts as a potent inhibitor of the IMD pathway in the larval trachea [44]. Overall, the sum of these effects on the performance of the IMD pathway should result in higher activities in the adult trachea, which is consistent with our observations with AMP reporter strains.

Mucins or mucin-like substances are of central importance for the functionality of airways. Their great relevance is supported by the fact that dysfunction of mucin secretion is associated with several different diseases, particularly of the lung and intestine [45]. In the larval trachea, almost no mucin gene expression was found, whereas numerous Muc and Mur coding genes [46] are expressed in the adult tracheal system. Mucins are of vital importance for the functionality of the vertebrate lung, where they act as part of the lining covering the airway epithelium [30,31]. There, they act as a protective lining and a structure that traps inhaled particles. Transport of the mucus allows for getting rid of these trapped particles [47]. Although we have no information about the role of mucins in the fly’s adult tracheal system, it appears that it resembles more features of the mammalian airway system than the larval tracheal system. Interestingly, the two most specific and abundant mucins, *Muc68D* and *Mur29B* take similar roles in the *Drosophila* intestine [48], where *Muc68D* was tightly associated with cystic fibrosis-like phenotype in this organ [48].

The expression of genes coding for metabolically relevant genes, which affect carbohydrate and lipid metabolism, also differed between larval and adult trachea. Among them, the increased expression of the two sugar transporters, *Tret1-1* and *Tret1-2,* is relevant [49,50]. The different members of the maltase family stand out, although their functional significance is still not understood. In the intestine, their expression depends on the composition of the diet [51]. Also elevated are genes coding for regulators such as *Tobi*, *Amy-d*, and *Amy-p*. These results imply that carbohydrate metabolism is more important in the adult trachea than in the larval trachea [52]. Concerning lipid metabolism, it is notable that two central genes involved in lipid catabolism, Brummer (*bmm*) and *Lip4*, are more abundant in the adult trachea [53,54], whereas *Lip1* appears to be specific to the larval trachea. The observation that *Tobi* and *Lip4* are predominantly expressed in adult trachea implies that changes in insulin signaling between adult and larval trachea are part of the differential transcription observed between both tissues [55].

Another aspect that deserves discussion is the difference concerning those genes whose products are responsible for the assembly of the chitinous structures of the trachea. Almost all gene families associated with these structures are affected by these differences. This observation is true for the enzymes that digest the extracellular matrix and thus enable growth processes, but also for the most diverse structure-giving proteins. Striking is the expression of the *Lcp* and *Ccp* genes, which is restricted to larvae. On the other hand, some chitinases and members of the *ldgf* family are more abundant in the adult tracheal system. This switch from the larval to the adult tracheal system is particularly striking for the largest of these families, the *Cpr* genes. Here, we have members that are almost exclusive to larvae; others are exclusive to adults. These differences may have several causes. The larval tracheal system is fully functional but still developing, whereas the adult tracheae are fully grown. In addition, the structures in adults are much more complex because elements are present there, such as the air sacs, that are not part of the larval tracheal system. Different autofluorescent properties in larval and adult trachea may reflect these differences. While 408/460 fluorescence is observed in both adults and larvae, 488/516 fluorescence appears to be restricted to adults. In addition, the differences between larval and adult trachea may also have other reasons. For example, larvae and adults have different ecological niches, which are also characterized by different oxygen concentrations. In addition, however, it should also be noted that adult flies actually fly and thus use what is probably the most energy-intensive form of locomotion [56].

The main components of the *Drosophila* circadian clock are all overrepresented in the adult tracheal system. This observation is true for all relevant circadian clock genes, including *per*, *tim*, *cry*, and *Clk*. Here, the two core components of the clock, *per*, and *tim*, are expressed several orders of magnitude higher in the adult trachea, implying that the adult trachea contains a peripheral clock that allows adjusting the performance of the respiratory system to the different needs during the day. However, the presence of a peripheral clock in the adult trachea has not been shown yet. Several organs of *Drosophila* contain a peripheral clock, including the intestine, the prothoracic gland, the malpighian tubules, and different sensory organs [57,58,59]. A peripheral circadian clock in the adult tracheal system would not be surprising simply because the adult respiratory system might benefit from circadian adjustments in its performance. In addition, peripheral clocks are operative in the mammalian lung [60,61].

Finally, it is necessary to discuss some technical aspects, advantages, disadvantages, and limitations of the study. The small amount of material that results from a clean, meaning low-contamination, preparation of adult trachea required the use of an amplification protocol. Here, we employed a whole cDNA amplification approach based on earlier work in the lab and on the smart-seq2 approach [62,63,64]. This approach certainly has advantages and disadvantages compared to single-cell sequencing. The latter has invaluable advantages with respect to cellular resolution. Our preferred approach does not provide a cellular resolution, but on the other hand, (1) gives us information about the specific transcript signatures of the whole organ and (2) has a much greater sequencing depth, allowing in-depth transcript analyses. Moreover, it must be kept in mind that, to date, no well-suited driver line that specifically targets the adult trachea is available. Even the canonical tracheal driver misses important regions of the adult trachea, including major branches in the thorax and the abdomen. The expression pattern of the other drivers tested, *emp-Gal4* and *flz-Gal4*, partially complements this, which must be kept in mind in studies aiming to target the entire adult tracheal system.

Concludingly these novel datasets should aid the further analyses of insect adult tracheal systems in general as they represent a superb resource and starting point for in-depth analyses aiming to understand the molecular and cellular basis of tracheal functionality.

## 4. Materials and Methods

### 4.1. Fly Stocks and Husbandry

Fly stocks were raised and cultured at 25 °C with 50–60% relative humidity in a 12:12 h day and night rhythm. For transcriptomic analyses, either *w^1118^* (BDSC #5905) female flies (5–7 days old) or 3rd instar larvae of mixed sex were used for dissection of the corresponding trachea. The *Btl-Gal4*, *UAS-GFP* fly line was a kind gift from the Maria Leptin Group (EMBL, Heidelberg, Germany). The *Tim-GFP* line was a kind gift from Philip Karpowicz (University of Windsor, Windsor, ON, Canada). The *emp*- (BDSC #66904), *flz*- (BDSC #67456), *geko*- (BDSC #66833), *ex*- (BDSC #76170), *Samuel-Gal4* (BDSC #76173), *Drs-GFP* (BDSC #55707) and *UAS-GFP* (BDSC #52262) flies were obtained from Bloomington *Drosophila* Stock Center.

### 4.2. Immunohistochemistry and Fluorescence Microscopy

The reporter line for Tim expression was stained with a primary α-GFP antibody from a mouse (DSHB, Iowa City, USA, 8H11) and a goat α-mouse antibody conjugated with Alexa Fluor 488 (Jackson ImmunoResearch, Cambridgeshire, UK, 115-545-062). Sagittal sections of the whole flies were stained with GFP Booster (ChromoTek, Martinsried, Germany, Catalogue number: gb2AF488), because the native GFP fluorescence was impaired during the fixation procedure. Antibody staining of larval and adult tracheae was visualized by an AxioImager Z1 equipped with an Apotome (Carl Zeiss Microscopy, Oberkochen, Germany) and with a Zeiss LSM880 (Carl Zeiss Microscopy, Oberkochen, Germany). To visualize the adult tracheal system, we have utilized the autofluorescence properties of tracheal epithelial cells. Head and thoracic tracheal cells were excited with 405 nm, and detection was conducted at 460 nm. The abdominal tracheal structures were excited with 488 nm and detected with a 516 nm laser. Emitted light was detected using the LSM 880 Airyscan detector with a 458/561 main beam splitter (Carl Zeiss Microscopy, Oberkochen, Germany). Airyscan processed images were selected for maximum intensity projection and digitally stitched together.

### 4.3. Dissection of Tracheae

Adult tracheae were dissected from five to seven-day-old adult flies. Dissection took place in phosphate-buffered saline by holding the flies with forceps (Dumont No. 5, Neolab, Heidelberg, Germany). Trachea were manually dissected by opening the carcass and removing as much as possible of the tracheal tissue from the head, thorax, and abdomen of three animals per replicate. Tracheal tissue can be easily distinguished from the remaining tissue by the white shiny color. The trachea were released from the forceps into the buffer. Due to buoyancy, the larger air sacs tend to float up. All tracheal parts were removed from the dissection buffer by careful pipetting. Excess buffer was removed with caution before up and down pipetting in RNA Magic. Larval tracheae were dissected from three animals per replicate. Here, we used mid-stage L3 larvae (prior to becoming late L3, wandering larvae). For material from whole flies, three whole flies were used. The dissected tissue and whole animals were suspended in RNA Magic (Bio Budget, Krefeld, Germany) and homogenized in a bead mill using 1.4 mm zirconia beads (Biolabproducts, Bebensee, Germany).

### 4.4. RNA Extraction, cDNA Synthesis and PCR Amplification

RNA isolation was performed using the Trizol method, as described before [25]. RNA concentration from whole animals was measured and diluted to 1.5 ng/µL before use to ensure equality for further processing of all samples. The cDNA synthesis and amplification of entire cDNAs was essentially performed as described earlier with slight modifications adjusted to a 10 µL reaction volume [62,63,65,66]. An oligo dT primer and a cap finder primer that binds the 5′-cap of mRNA were used. With this cDNA synthesis, every single transcript carried a tag and could be amplified with one universal primer, independent of the sequence or length of the fragment. For amplification, LA Taq polymerase (Takara Bio Inc., Saint-Germain-en-Laye, France) was used with 2.5 µL cDNA per 35 µL amplification reaction. After optimization of the cycling protocol to ensure being always in the exponential phase of amplification, the corresponding parameters were used. Cycling conditions were set up as follows: 94 °C 1 min, 98 °C 10 s, 68 °C 5 min, 72 °C 10 min. Steps two and three were cycled for the previously determined number of cycles. Excessive primers, nucleotides, salts, and enzymes were removed with Monarch PCR & DNA cleanup Kit (New England Biolabs GmbH, Frankfurt, Germany) by following the manufacturer’s protocol. DNA concentrations were determined in a Qubit using the high-sensitivity dsDNA kit (ThermoFisher, Karlsruhe, Germany). Statistical significance in qRT-PCR was evaluated by Mann-Whitney Test.

### 4.5. Transcriptomic Analyses

Library preparation was performed using a protocol adapted from Picelli et al. 2014. The samples were equimolar pooled and quantified (with Agilent Bioanalyzer, DNA 7500). Sequencing was performed on the NextSeq 500 (Illumina, Berlin, Germany) using the High Output Kit v2.5 (75 cycles). The statistical model uses a separate Generalized Linear Model (GLM) for each gene, assuming that the read counts follow a Negative Binominal distribution. Differential expression was tested due to group and compared against a control group (Wald Test). Whether genes are differentially expressed was indicated with asterisks in the graphs that compare RPKM values.

Differential expression of sequencing data was performed with the CLC genomic workbench software (Qiagen, Hilden, Germany). In brief, reads were normalized and mapped to the BDGP6 reference by the RNA-Seq Analysis tool. Differential expression was calculated with the Differential Expression for RNA-Seq tool. The GO analysis was conducted by using the g:Profiler tool. These GOs were clustered using the Cytoscape software, employing Enrichment Map and Auto Annotate plug-ins.

### 4.6. Synchrotron-Radiation Based X-ray Computed Micro-Tomography

Flies were imaged using synchrotron radiation-based micro-computed tomography (SRµCT) at the Imaging Beamline P05 (IBL) [67] operated by the Helmholtz-Zentrum Hereon at the storage ring PETRA III (Deutsches Elektronen Synchrotron–DESY, Hamburg, Germany). A photon energy of 30 keV and a sample-to-detector distance of 100 mm has been used for imaging. Projections were recorded using a commercial 50 MP CMOS camera system with an effective pixel size of 0.46 µm. For each tomographic scan, 1501 projections at equal intervals between 0 and π were recorded. Tomographic reconstruction was conducted by applying a transport of intensity phase retrieval approach and using the filtered back projection algorithm (FBP) implemented in a custom reconstruction pipeline [68] using Matlab (Math-Works) and the Astra Toolbox [69]. For further processing, raw projections were binned two times, resulting in an effective pixel size of the reconstructed volume of 0.92 µm.

## Figures and Tables

**Figure 1 ijms-24-05628-f001:**
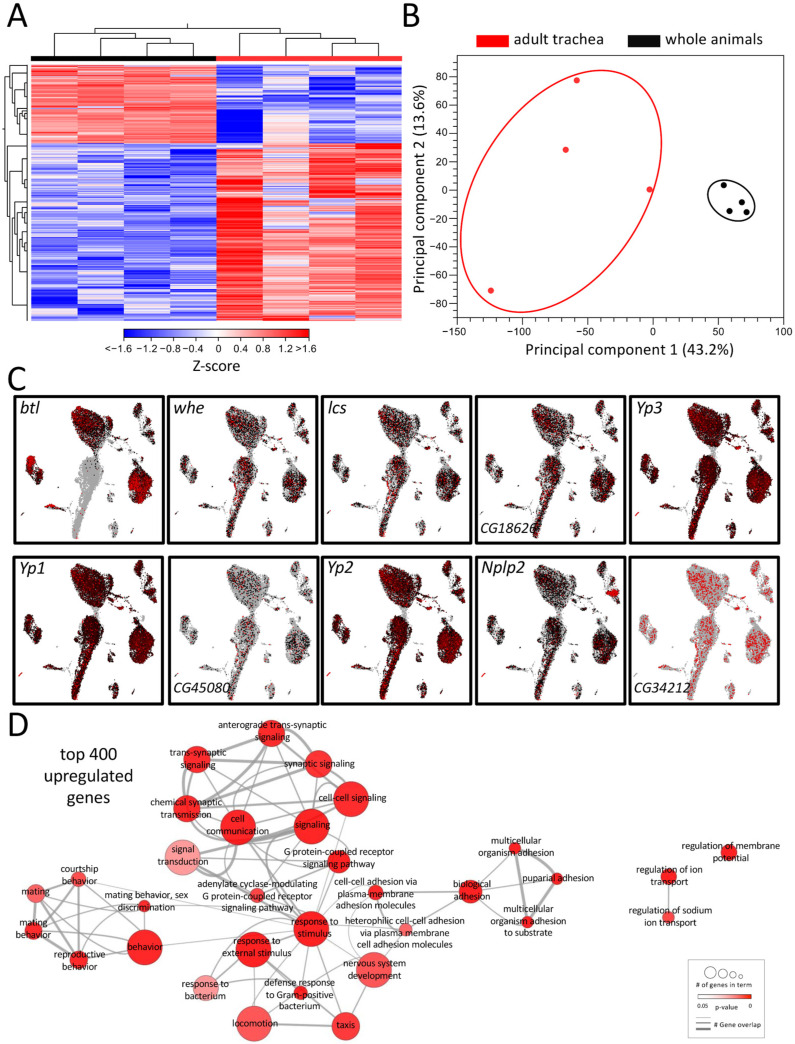
Transcriptomic analyses of adult respiratory epithelia compared to whole flies. (**A**) Heatmap of all differentially expressed genes (DEGs) between adult trachea (red) and whole flies (black). DEG cutoff: fold-change > 1.5. (**B**) PCA of all replicates from adult trachea and whole flies. Circles were added manually. (**C**) Transcript plots of selected genes showing high abundances in the RNAseq experiments based on the *Drosophila* single-cell atlas https://scope.aertslab.org/#/FlyCellAtlas/FlyCellAtlas%2Fs_fca_biohub_trachea_10x_ss2.loom/gene, accessed 12 December 2022 [28]. (**D**) GO analysis with the top 400 upregulated DEGs with a cutoff value of FDR < 0.05. Node size represents the number of associated genes in the GO term. The node color represents the *p*-value. Edge appearance represents the number of shared genes between the GO terms.

**Figure 2 ijms-24-05628-f002:**
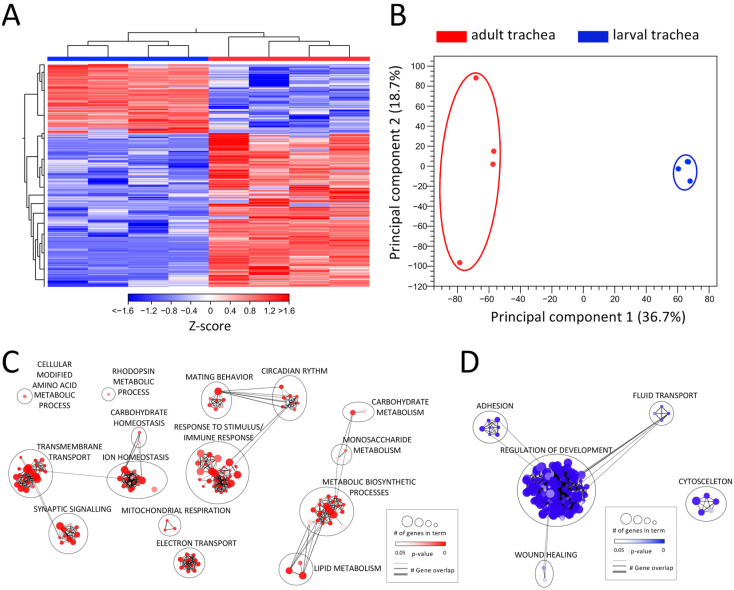
Comparing the transcriptomic signatures of adult and larval trachea. (**A**) Heatmap of all DEGs between adult tracheal (red) and larval trachea (blue). (**B**) PCA of all replicates from adult trachea and larval trachea. Circles were added manually. (**C**,**D**) GO analysis with the upregulated (**C**) and downregulated (**D**) DEGs with a cutoff value of FDR < 0.05. Node size represents the number of associated genes in the GO term. The node color represents the *p*-value. Edge appearance represents the number of shared genes between the GO terms.

**Figure 3 ijms-24-05628-f003:**
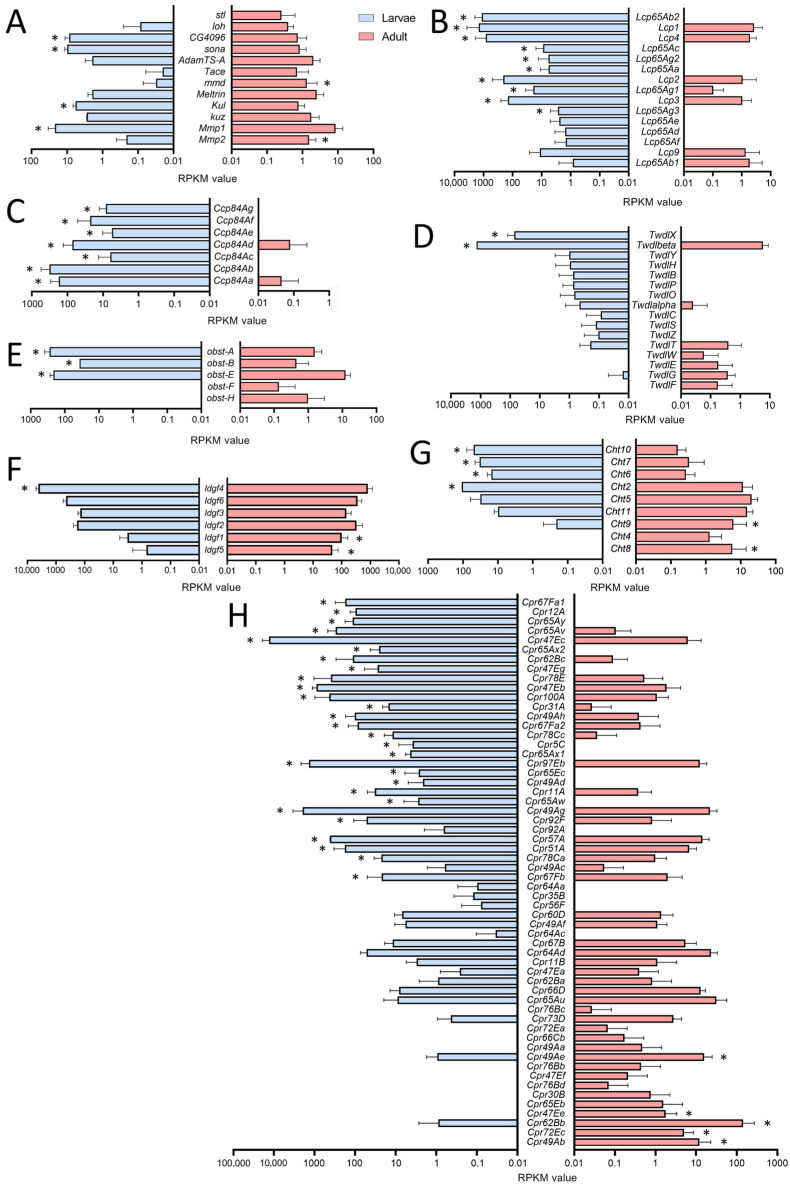
Cuticle-, chitin- and ECM-associated genes in larval and adult trachea. (**A**–**H**) Bar charts showing the Reads per kilobase of transcript per million mapped reads (RPKM) values compared between larval (blue) and adult trachea (red). In (**A**), genes coding for metalloproteinase; in (**B**), those coding for Larval Cuticle Proteins (*Lcp*); and in (**C**), those of the *Ccp* family are shown. (**D**) shows members of the Tweedle (*Twdl*) family, (**E**) those of the obstructor (*obst*) group, (**F**) members of the imaginal disc growth factor (*ldgf*) family, (**G**) those of the Chitinase (*Cht*) family, and (**H**) members of the cuticular protein (*Cpr*) group. Asterisks indicate a differential gene expression based on the RNA-Seq analysis. Bars show mean with SD, n = 4.

**Figure 4 ijms-24-05628-f004:**
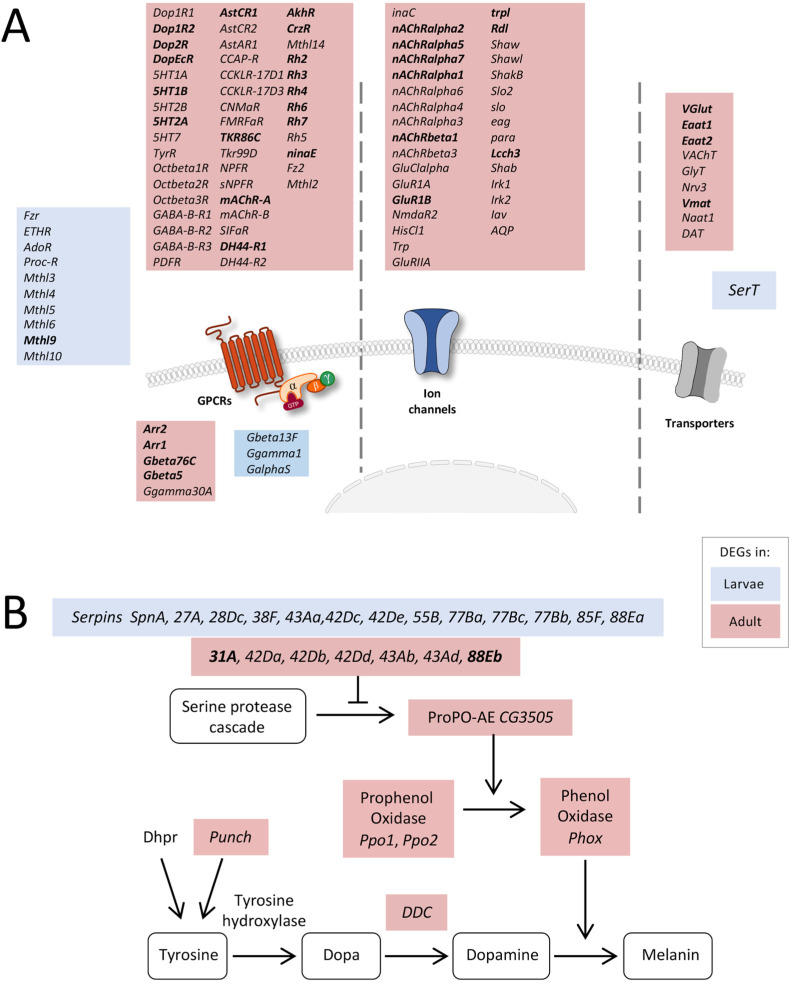
Expression of receptor-, transporter-, and transmembrane channel-coding genes. (**A**) Schematic overview of differentially expressed genes coding for proteins involved in signal transduction in adult and larval trachea. Receptors, channels, and transporters labeled in red represent the upregulation of the corresponding DEGs in adult tracheal cells, whereas blue-labeled ones represent upregulation in the larval system. (**B**) Scheme of components involved in the melanization process of larval and adult tracheal cells. Shown are all components of the melanization response in *Drosophila* and genes coding for corresponding components that were significantly upregulated either in larval (blue) or adult (red) trachea. Based on DEG with an FDR cutoff value of *p* < 0.05. DEG with a fold change >100 are emphasized in bold.

**Figure 5 ijms-24-05628-f005:**
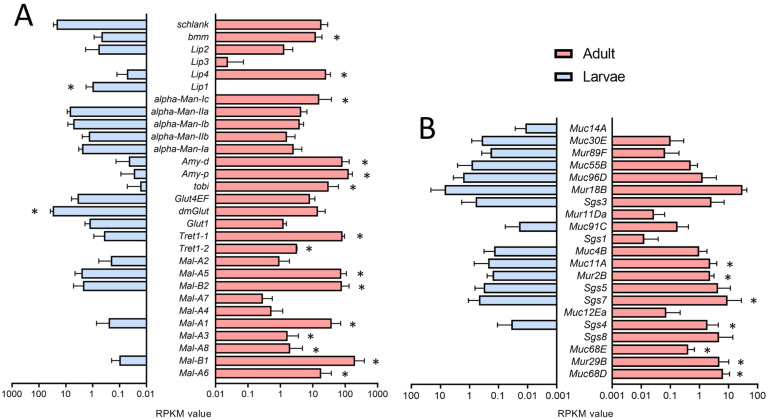
Expression of genes associated with metabolism and production of mucin-like products. (**A**,**B**) Bar charts showing the RPKM values compared between larval (blue) and adult trachea (red). Expression levels of genes that are associated with lipid metabolism (**A**) and *Muc*, *Mur,* and *Sgs* genes (**B**) that are differentially transcribed between larvae (blue) and adults (red). Asterisks indicate a differential gene expression based on the RNA-Seq analysis. Bars show mean with SD, n = 4.

**Figure 6 ijms-24-05628-f006:**
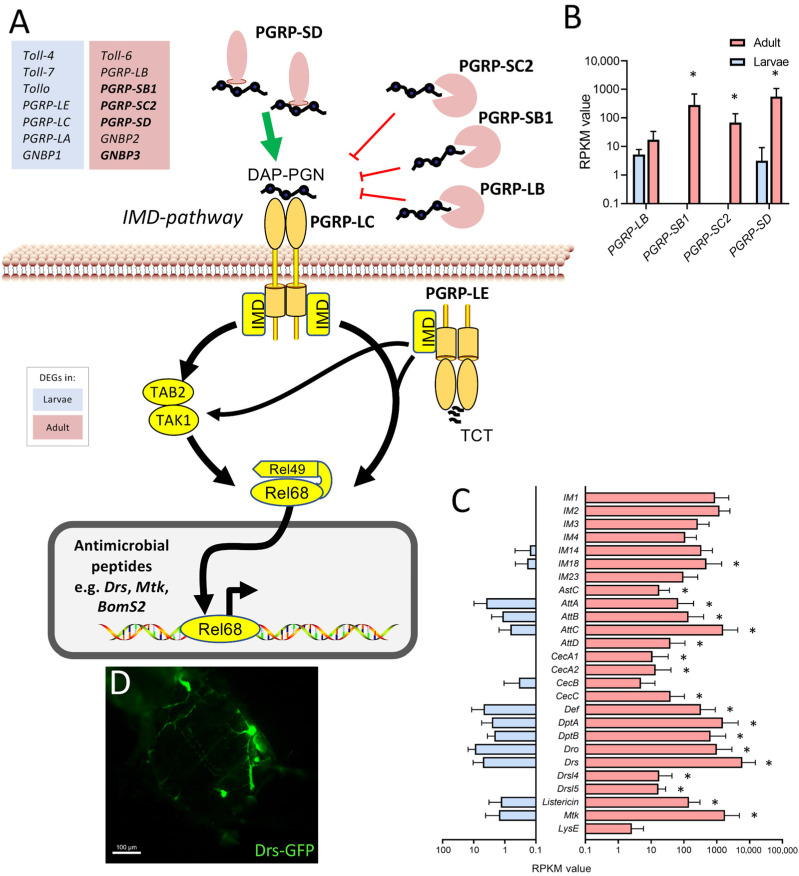
Components of the immune system expressed at higher levels in the adult trachea. (**A**) Schematic overview of the extra- and intracellular components of the IMD pathway. (**B**) RPKM expression values of different PGRPs compared between larval (blue) and adult trachea (red). (**C**) RPKM expression values of different antimicrobial molecules compared between larval and adult trachea. Asterisks indicate a differential gene expression based on the RNA-Seq analysis. Bars show mean with SD, n = 4. (**D**) *Drs*-GFP expression in the trachea of the adult abdomen.

**Figure 7 ijms-24-05628-f007:**
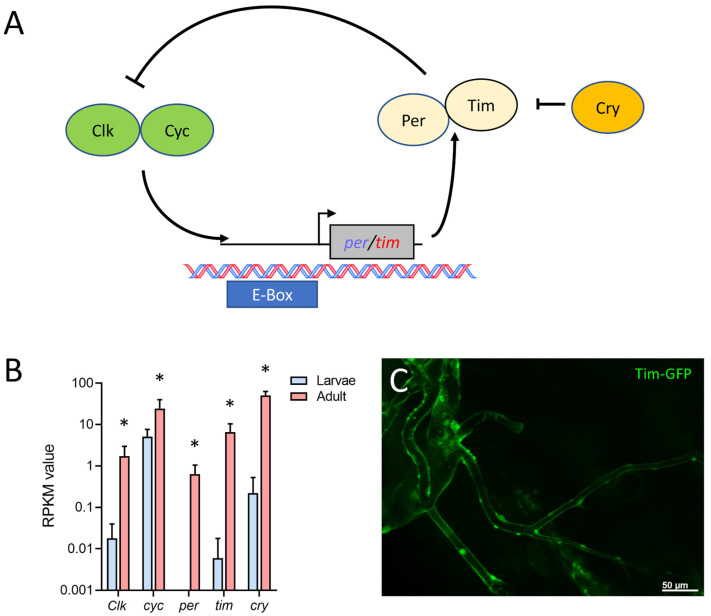
Adult tracheal cells express core components of the circadian clock. (**A**) Schematic overview of the circadian clock in *Drosophila*. (**B**) Normalized read count (RPKM) values of clock coding genes for *Clock* (*Clk*), *cycle* (*cyc*), *period* (*per*), *timeless* (*tim*) and *cryptochrome* (*cry*). Bars show mean with SD, n = 4. Asterisks indicate a differential gene expression based on the RNA-Seq analysis. (**C**) Microscopic confirmation of in vivo Tim expression in abdominal adult trachea.

**Figure 8 ijms-24-05628-f008:**
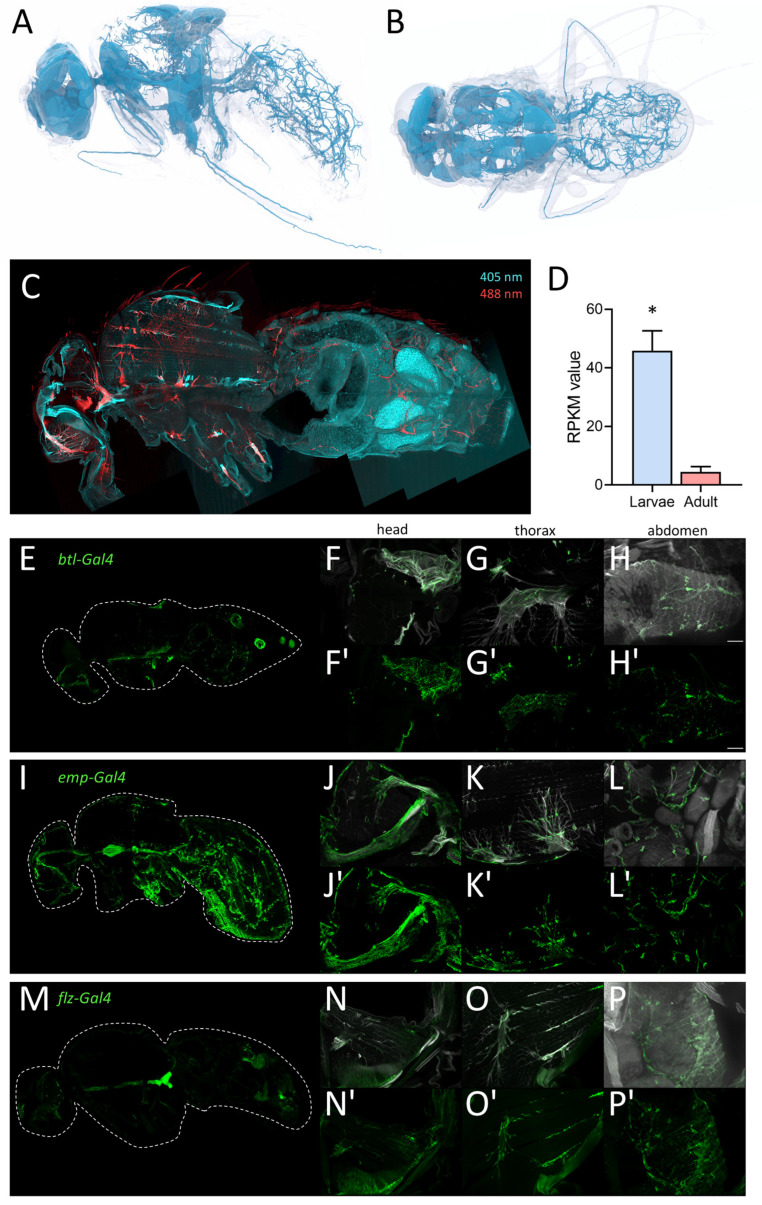
Morphology of the tracheal system in adult animals. (**A**,**B**) Structure of the tracheal system of adult flies as evaluated by micro-CT analysis, lateral (**A**) and dorsal view (**B**). To visualize tracheal structures in adults, we also employed autofluorescence (**C**). Here, longitudinal sections were analyzed with a confocal laser scanning microscope. The tracheal cells were excited by 405 nm (cyan; detection 460 nm) and 488 nm (red, detection 516 nm) lasers and detected for corresponding autofluorescence of the trachea in the head, thorax, and abdomen (**C**). (**D**) RPKM values of larval (blue) and adult (red) trachea for the canonical trachea driver gene *btl*. Asterisk indicates a differential gene expression based on the RNA-Seq analysis. (**E**–**P’**) Expression of tracheal specific Gal4 driver lines in whole fly, head, thorax, and abdomen. Sagittal sections of *btl-Gal4* (**E**–**H’**), *emp-Gal4* (**I**–**L’**) and *flz-Gal4* (**M**–**P’**) crossed to *UAS-GFP* were stained with an anti-GFP nanobody. Tracheal air sacs and branches were simultaneously visualized with UV light (white; **F**–**H**, **J**–**L**, **N**–**P**). Detailed magnification of the head (**F**,**F’**,**J**,**J’**,**N**,**N’**), thorax (**G**,**G’**,**K**,**K’**,**O**,**O’**) and abdomen (**H**,**H’**,**L**,**L’**,**P**,**P’**) are shown. Scale = 50 µm.

## Data Availability

Primary data of this study have been deposited at the GEO database: GSE225000.

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
