# Peer review of "Adult and Larval Tracheal Systems Exhibit Different Molecular Architectures in Drosophila"

_ijms, 2023, doi:10.3390/ijms24065628_

Round 1

Reviewer 1 Report

The manuscript by Bossen et al contributes to the molecular characterization of the Drosophila adult tracheal system. By comparing the transcriptomes of larval and adult trachea the authors identify several genes and pathways that are differentially expressed and are most likely related to functional differences between the larval and adult trachea. The authors revisit the morphology of the adult tracheal system using micro-CT analysis and investigate the pattern of expression driven by tracheal Gal4 drivers in adults. The results are clearly presented and discussed. The experimental approach undertaken by the authors, namely, generating adult tracheal transcriptomes from dissected adult trachea, is technically challenging and the limitations of the study have been clearly presented. Not too many molecular studies are available for the Drosophila adult tracheal system and therefore this study is an important resource to those working in the field.

Specific points

Page 7, paragraph starting in line 199.

Please check if Meltrin, mmd and Tace are specifically found in the adult trachea. The graph shown in figure 3A seems not to entirely agree with the text.

Page 9, lines 239 to 246

The authors comment on the autofluorescence in the adult trachea detected at two different excitation wavelengths (405 nm and 488nm). Whereas the autofluorescence after excitation with 405 nm is observed both in the larval and adult tracheal system, the autofluorescence after excitation with 488 nm is restricted to parts of the adult thorax and head. This point is presented in the results session but is not further discussed. Do the authors have any suggestion regarding the basis of these differences? Could they be indicating functional differences in the tracheal system? Is there an overlap between the two autofluorescence patterns?

Discussion

At the beginning of the discussion (line 367) the authors attribute the transcriptome differences between the adult and larval trachea to differences in structure, physiology and cell types. Even though under lab conditions the Drosophila larvae and adults live in vials or bottles, a very controlled environment, in nature the larvae move within the substrate, whereas the adults are free to fly and disperse. Could the observed differences between the trachea of larvae and adult could also be correlated to the biology of larvae and adults in nature? Perhaps the authors might comment on this and if they think is relevant, include a short discussion about this point in the discussion?

Material and Methods

It is not clear why the authors employ immunohistochemistry nor a GFP Booster since the lines express GFP. Are the GFP expression levels too low? A short explanation for this point should be included in session 4.2 (line 481). Additionally, the catalog number of the antibodies should also be provided.

Dissecting Drosophila adult trachea is not a trivial task and the authors should give more details about how it was actually done (line 496). Where the flies pinned for dissection? Were they dissected in saline? How was the tissue stored prior to RNA extraction?

The authors should provide information regarding the statistical analyses.

Author Response

Response to Reviewer 1 comments:

The manuscript by Bossen et al contributes to the molecular characterization of the Drosophila adult tracheal system. By comparing the transcriptomes of larval and adult trachea the authors identify several genes and pathways that are differentially expressed and are most likely related to functional differences between the larval and adult trachea. The authors revisit the morphology of the adult tracheal system using micro-CT analysis and investigate the pattern of expression driven by tracheal Gal4 drivers in adults. The results are clearly presented and discussed. The experimental approach undertaken by the authors, namely, generating adult tracheal transcriptomes from dissected adult trachea, is technically challenging and the limitations of the study have been clearly presented. Not too many molecular studies are available for the Drosophila adult tracheal system and therefore this study is an important resource to those working in the field. 

Specific points

Page 7, paragraph starting in line 199.

Please check if Meltrin, mmd and Tace are specifically found in the adult trachea. The graph shown in figure 3A seems not to entirely agree with the text.

We would like to thank you for that - we have changed it accordingly.

Page 9, lines 239 to 246

The authors comment on the autofluorescence in the adult trachea detected at two different excitation wavelengths (405 nm and 488nm). Whereas the autofluorescence after excitation with 405 nm is observed both in the larval and adult tracheal system, the autofluorescence after excitation with 488 nm is restricted to parts of the adult thorax and head. This point is presented in the results session but is not further discussed. Do the authors have any suggestion regarding the basis of these differences? Could they be indicating functional differences in the tracheal system? Is there an overlap between the two autofluorescence patterns?

We described this in more detail in the results and the Material and methods section and included a short discussion of the implications in the discussion section.

Discussion

At the beginning of the discussion (line 367) the authors attribute the transcriptome differences between the adult and larval trachea to differences in structure, physiology and cell types. Even though under lab conditions the Drosophila larvae and adults live in vials or bottles, a very controlled environment, in nature the larvae move within the substrate, whereas the adults are free to fly and disperse. Could the observed differences between the trachea of larvae and adult could also be correlated to the biology of larvae and adults in nature? Perhaps the authors might comment on this and if they think is relevant, include a short discussion about this point in the discussion?

We added a short description of this to the introduction and a longer discussion to the discussion section to highlight the differences between larvae and adults.

Material and Methods

It is not clear why the authors employ immunohistochemistry nor a GFP Booster since the lines express GFP. Are the GFP expression levels too low? A short explanation for this point should be included in session 4.2 (line 481). Additionally, the catalog number of the antibodies should also be provided.

We described this now in the Material and methods section. It was necessary, because the fixation required for the sectioning led to strongly reduced GFP signals.

Dissecting Drosophila adult trachea is not a trivial task and the authors should give more details about how it was actually done (line 496). Where the flies pinned for dissection? Were they dissected in saline? How was the tissue stored prior to RNA extraction? 

We now added a detailed description if the dissection procedure to the Material and methods section.

The authors should provide information regarding the statistical analyses.

This has been added to the corresponding subsection in the Material and methods part of the manuscript.

Reviewer 2 Report

Gene expression differences between the larval and adult trachea is an important question to address and the data can be very useful. Similarly, careful analysis of GAL4 tools that can be used to visualize and drive expression of RNAi and overexpression constructs is helpful to the community. I am nevertheless somewhat concerned with aspects of experimental design and how the results are interpreted. Specifically, a critical difference between the larval and adult trachea is that the larval trachea is a developing structure. There is considerable growth over the larval period that is punctuated by molts that degrade the apical cuticle. Ecdysone signaling before and during the molts have an enormous effect on gene expression. Can you be more specific on the age of the adults that you used for tissue extraction and what stage you used for the larvae. Where all the larvae of the same developmental stage? You mentioned that you used females for the adult tissue, did you use female larvae as well to control for sex differences in expression? The intro and discussion should address these issues. The cuticle and chitin (and proteinase) gene expression differences make a lot more sense in the light of molting and remodeling. Likewise, the need for substantial larval growth likely affects the genes used for metabolism.

It is interesting that the microCT analysis of the adult trachea shows more complexity than the GFP expression from btl-GAL4. Is this really an adult phenomenon? Can you compare the microCT of larval trachea with the btl>GFP to show how well these two tools compare in the larvae. The autofluorescence in adult tissues didn’t seem to be very complete compared to microCT. I am not sure what the value of adding that data to the paper (especially where you placed it in Figure 4).

I liked the way you displaced the expression differences in a log scale in Figures 4, 6 and 7 for example. Can you do the same for the data for Figure 5? I would also suggest having a threshold for expression (FPKM). Some of the genes you show have FPKM values well under 1 in the highest of the two samples. They may be significantly different, but if they are not really expressed then it is probably not very meaningful.

Lines 67-69. I am not sure why it makes sense that the more complex architecture of the adult organ would result in expression differences. It seems more likely that developmental factors are more relevant.

Author Response

Reply to Reviewer 2 comments:

Gene expression differences between the larval and adult trachea is an important question to address and the data can be very useful. Similarly, careful analysis of GAL4 tools that can be used to visualize and drive expression of RNAi and overexpression constructs is helpful to the community. I am nevertheless somewhat concerned with aspects of experimental design and how the results are interpreted. Specifically, a critical difference between the larval and adult trachea is that the larval trachea is a developing structure. There is considerable growth over the larval period that is punctuated by molts that degrade the apical cuticle. Ecdysone signaling before and during the molts have an enormous effect on gene expression. Can you be more specific on the age of the adults that you used for tissue extraction and what stage you used for the larvae. Where all the larvae of the same developmental stage? You mentioned that you used females for the adult tissue, did you use female larvae as well to control for sex differences in expression? The intro and discussion should address these issues. The cuticle and chitin (and proteinase) gene expression differences make a lot more sense in the light of molting and remodeling. Likewise, the need for substantial larval growth likely affects the genes used for metabolism.

This aspect is now discussed in the discussion section and we also added a short statement in the introduction that highlights the differences between larval and adult trachea.

The age of the adults and sex of the larvae are now described in the Material and methods section.

It is interesting that the microCT analysis of the adult trachea shows more complexity than the GFP expression from btl-GAL4. Is this really an adult phenomenon? Can you compare the microCT of larval trachea with the btl>GFP to show how well these two tools compare in the larvae. The autofluorescence in adult tissues didn’t seem to be very complete compared to microCT. I am not sure what the value of adding that data to the paper (especially where you placed it in Figure 4).

This is really an interesting point. We only have the microCT data from the adult tracheal system, which was our main aim. The differences are indeed striking, and we aimed to focus on this in the last figure, where we used different driver lines that all mark slightly different parts of the adult tracheal system. For larvae, we do not see such a discrepancy. It is easy to control, as the entire tracheal system can be visualized without using microCT. We added this (and now discussed this), because it points to structural differences as the second autofluorescence is almost completely absent from the larval trachea.

I liked the way you displaced the expression differences in a log scale in Figures 4, 6 and 7 for example. Can you do the same for the data for Figure 5? I would also suggest having a threshold for expression (FPKM). Some of the genes you show have FPKM values well under 1 in the highest of the two samples. They may be significantly different, but if they are not really expressed then it is probably not very meaningful.

We thought about this, but we believe that displaying the data for Figure 5 in the way we did is better suited to convey the relevant information. While the log scale type of display is ideally suited to show multiple members of a gene family, especially when the particular members of the family are not per se of greatest interest. In the case of figure 5, the situation is different, and for each of the genes that are listed a large amount of information is available. This, we think that it might be easier for interested readers to grasp the relevant information. With respect to the low FPKM values, you are absolutely right, but we decided to keep them in the corresponding figures to give a complete picture about the expression of all (or most) gene family members.

Lines 67-69. I am not sure why it makes sense that the more complex architecture of the adult organ would result in expression differences. It seems more likely that developmental factors are more relevant.

We discussed this in more detail and added additional explanations to the discussion section.

Round 2

Reviewer 2 Report

I was concerned that the larval samples may not have been closely staged. To some extent that can be problematic since you would find a significant number of differentially expressed genes if you chose a time point in the middle of the 3rd instar when the larvae are starting to wander compared to late in the stage when ecdysone signaling is strongly rising. Nevertheless, comparing a mixed larval sample with an adult sample does have a lot of merit.  I think it would be important to mention this in the last part of the discussion when you discuss the limitations of the study. You could mention that you might find more differences if you timed the larval stages relative to molting and pupariation and that that might be a useful future direction. 

I accept that you don't want to do the microCT on the larval stages, and believe that breathless probably is a very strong indicator of trachea in the larvae. Your attempts to define the adult trachea with breathless, the new Gal4 lines you use and the autofluorescence are very interesting, but are too disjointed the way they are currently presented. It doesn't make a lot of sense to have the autofluorescence in figure 4. I would recommend making a single figure showing the microCT, the autofluorescence and all the GAL4 lines and then talking about them all in one paragraph or section rather than in 3 different places. 

Author Response

I was concerned that the larval samples may not have been closely staged. To some extent that can be problematic since you would find a significant number of differentially expressed genes if you chose a time point in the middle of the 3rd instar when the larvae are starting to wander compared to late in the stage when ecdysone signaling is strongly rising. Nevertheless, comparing a mixed larval sample with an adult sample does have a lot of merit.  I think it would be important to mention this in the last part of the discussion when you discuss the limitations of the study. You could mention that you might find more differences if you timed the larval stages relative to molting and pupariation and that that might be a useful future direction. 

Thank you for this remark. We clarified the sampling procedure of L3 larvae in the Material and Methods section and now explained that we explicitely only took mid-stage L3 larvae, before they enter the late, wandering larvae stage.

I accept that you don't want to do the microCT on the larval stages, and believe that breathless probably is a very strong indicator of trachea in the larvae. Your attempts to define the adult trachea with breathless, the new Gal4 lines you use and the autofluorescence are very interesting, but are too disjointed the way they are currently presented. It doesn't make a lot of sense to have the autofluorescence in figure 4. I would recommend making a single figure showing the microCT, the autofluorescence and all the GAL4 lines and then talking about them all in one paragraph or section rather than in 3 different places. 

You are completely right. We combined now the three figures as figure 8. Consequently, we deleted figure one and replaced the autofluorescence analysis from figure 4 (now figure 3) to figure 8. Thus, we have a comprehensive description of the different driver lines in relation to the structure of the trachea.